# Head Injury and Amyotrophic Lateral Sclerosis: Population-Based Study from the National ALS Registry

**DOI:** 10.3390/brainsci15020143

**Published:** 2025-01-31

**Authors:** Jaime Raymond, Ileana M. Howard, Jasmine Berry, Theodore Larson, D. Kevin Horton, Paul Mehta

**Affiliations:** 1Agency for Toxic Substances and Disease Registry/Centers for Disease Control and Prevention, Atlanta, GA 30341, USA; thy7@cdc.gov (J.B.); thl3@cdc.gov (T.L.); dth9@cdc.gov (D.K.H.); pum4@cdc.gov (P.M.); 2Rehabilitation Care Services, VA Puget Sound Healthcare System, Seattle, WA 98108, USA; ileana.howard@va.gov; 3Department of Rehabilitation Medicine, University of Washington, Seattle, WA 98195, USA

**Keywords:** amyotrophic lateral sclerosis, head injury, age of diagnosis, traumatic brain injury, epidemiology

## Abstract

**Background/Objectives:** To examine if head injury (HI) is associated with age at ALS diagnosis in the United States. **Methods:** In this cross-sectional populationf-based analysis, we identified patients with ALS who were registered from 2015 to 2023 who completed the Registry’s head trauma survey module. The association between HI and age at ALS diagnosis was assessed using multivariate analysis. **Results:** Of the 3424 respondents, 56.6% had experienced a HI. The adjusted odds ratio (aOR) for an ALS diagnosis before age 60 years for patients with a HI was 1.24 (95% CI, 1.07–1.45). One or two HIs had an aOR of 1.15 (95% CI, 0.97–1.36), and five or more HIs had an aOR of 1.58 (95% CI, 1.19–2.09). HI before age 18 years yielded an aOR of 2.03 (95% CI, 1.53–2.70) as well as HI between the ages of 18 and 30 years (aOR = 1.48, 95% CI: 1.06–2.06)). When narrowing the analysis to patients with HI before age 18 compared with patients with no HI, we found an association with HI that led to an emergency department or hospital visit (aOR = 1.50 (95% CI: 1.21–1.86)). **Conclusions:** In this cross-sectional analysis of ALS patients, HIs occurring in childhood and early adulthood and the number of HIs increased the odds of being diagnosed before age 60 years. These results suggest that HI continues to be a risk factor for ALS and could be associated with a younger age of diagnosis.

## 1. Introduction

Amyotrophic lateral sclerosis (ALS) is a neurodegenerative disease that leads to progressive limb and bulbar weakness, respiratory impairment, and death, generally within three to five years from the onset of symptoms. The incidence of ALS in the United States [1] is estimated at one to two per 100,000 person-years, and the lifetime risk is estimated to be one in 400 persons [2]. The average age of onset is age 58–63 years for sporadic ALS cases [3,4], with a diagnosis occurring approximately nine to 14 months after symptom onset [5]. There is great interest in identifying potential risk factors for ALS [6]. The current understanding of risk factors include male sex, smoking, and military service. Other potential risk factors include exposure to heavy metals, pesticides, high BMI and nutritional state, β-N-methylamino-l-alanine (BMAA) [7], high levels of physical activity [8,9,10], and head injury (HI) [11].

HI refers to any transfer of mechanical injury to the head through external forces and exists along a spectrum of severity. When accompanied by specific clinical signs, symptoms, and laboratory or radiographic findings, HI may meet the criteria for a diagnosis of traumatic brain injury (TBI) [12]. Previous work suggests a possible association between history of TBI [12], especially moderate or severe injury, and increased risk of neurodegenerative diseases such as dementia and Parkinson’s disease [11,12]. TBI is thought to trigger neurodegeneration through a cascade of oxidative stress, dysregulated proteostasis, neuroinflammation [13], and TDP-43 mislocalization [14]. While the association between mild TBI and neurodegenerative disease is less strong, there is evidence suggesting that repetitive mild TBI is associated with an increased risk for neurodegenerative diseases such as chronic traumatic encephalopathy (CTE) [15]. For the purposes of this paper, HI will be used as a general term to refer to an exposure consisting of any trauma to the head region.

Previous studies have explored the association between HI and ALS, but the relationship remains inconclusive. Due to the higher incidence of ALS in military veterans, raising concerns about occupational exposure to HI, several studies have also focused on TBI in active-duty service members and veterans [16,17,18,19]. Meta-analyses in 2007 [20], 2017 [21], and 2021 [22] have summarized earlier epidemiological reports on this association. In addition, there is interest in histopathological overlap with accumulations of TAR DNA-binding protein 43 (TDP-43) and cytosolic accumulations in both HI and ALS [23]. An animal model involving SOD1 mice found early onset of ALS with repetitive mild HI [24], suggesting potential interplay between genetic risk factors and environmental triggers. Despite the growing evidence suggesting an association between HI and ALS, some studies have reported conflicting results. A meta-analysis conducted by Perry et al. found no significant association between HI and ALS risk in four studies [25]. Questions have also been raised about “reverse causation” due to the often-short time interval between HI and the diagnosis of ALS [21]. It is possible that early undiagnosed weakness from ALS could be the cause rather than the effect of HIs [21]. The meta-analyses conducted by Watanabe and Watanabe concluded that the association between ALS and TBI weakened when excluding cases with very short time lag (<1 year) between events [16]. Another meta-analysis reached similar conclusions in support of reverse causation, reporting a strong association between ALS risk and HIs that occurred for less than one year, while the association weakened when the time lag was set at fixed intervals greater than one year [22].

Due in part to the increasing awareness of late sequelae of HI, there has been an increase in research on the identification and prevention of sports-related and youth concussions [23]. Postmortem examinations of youth contact athletes who died before age 30 years [26] have shown evidence of CTE. Previous studies have examined the association between youth or nonprofessional athletics and a diagnosis of ALS [27,28,29], but the correlation appears to be weaker than with professional athletics [30]. A systematic review found no conclusive evidence relating repetitive sports-related concussion in amateur athletes to a diagnosis of neurological disease [31]. The aim of this data analysis was to examine whether HI is associated with age at ALS diagnosis in the United States based on analyzed survey data from the National ALS Registry web portal.

## 2. Materials and Methods

### 2.1. Analytic Population

In 2008, Congress authorized the establishment of the National ALS Registry (Registry) [32] through the federal Agency for Toxic Substances and Disease Registry (ATSDR), an environmental health agency administratively linked to the Centers for Disease Control and Prevention (CDC) [33]. The Registry was tasked with identifying epidemiologic trends, examining environmental and occupational factors that could be associated with ALS, and determining the disease’s public health burden [1]. While details about the Registry’s objectives are presented elsewhere [34], briefly, the Registry’s purpose is to quantify the incidence and prevalence of ALS in the United States, describe the patient demographics, and examine potential risk factors [35]. In October 2010, ATSDR launched the Registry [1]. To verify the ALS status of people who enroll online to the Registry, ATSDR uses six questions from the U.S. Department of Veterans Affairs ALS registry, which are reliable indicators for accurate ALS diagnoses [36].

Persons enrolling online can complete exposure modules that currently consist of 18 surveys related to possible risk factors for ALS (e.g., environmental exposure, job history, physical activity, pesticide use, head and neck injuries) [37]. These surveys were designed by the ALS Consortium of Epidemiologic Studies [38] at Stanford University [39,40] and do not require a healthcare provider’s help to complete [41]. To date, more than 100,000 surveys have been completed, representing the largest, most geographically diverse collection of ALS risk factor data available.

The Registry’s head trauma survey module launched on December 1, 2014. Participants are determined to have head trauma via the following question: “Have you ever had an injury to your head or neck?”. The module examines all types of HIs including severity and how the HI occurred throughout their lifetime. The survey contains eight questions assessing HIs prior to ALS diagnosis and covers the number of HIs, severity, and when and how the HI occurred (car crash, fall, fight, and explosion). Patients who had registered before the survey launched were able to log back in and take the survey. If the patient did not respond to a question about military history, for example, they would show missing information for that variable. Therefore, this analysis included HI data completed by patients between 1 December 2014 and 31 December.

### 2.2. Outcomes

The main predictor variable for this analysis was the self-report of a HI (yes/no). The primary outcomes were: age at ALS diagnosis, number of HIs, age at first HI, loss of consciousness (yes/no), time spent unconscious (no loss of consciousness, <5 min, 5–59 min, 1–24 h), HI requiring emergency department or hospitalization (yes/no), HI causing a skull fracture (yes/no), HI causing a seizure (yes/no), and HI causing memory loss (yes/no). We used prior to 60 years to define early diagnosis—ALS can affect people at any age, but sporadic cases typically start around then [42]. A secondary outcome included patients with ALS younger than 60 years who had a HI before the age of 18.

### 2.3. Covariates

Selected demographic characteristics including race, age at diagnosis, body mass index (BMI), military status (yes/no), and physical activity level not related to their occupation (never vigorous activity/vigorous activity) were abstracted for those who completed the HI survey module. Because of the high percentage of persons who were White, the race was coded as “White”, “Black”, or “Other”. If participants selected more than one race, we categorized them as “Other”. BMI was calculated with the standard formula: BMI = weight (lb)/[height (in)] 2 × 703 [43].

### 2.4. Statistical Analysis

We analyzed diagnosis before age 60 years versus at or after age 60 years. We used adjusted logistic regression models to estimate the odds ratios (ORs) and 95% confidence intervals [44] for multivariate analysis. Backward elimination was used to establish the final reduced logistic regression model. We included variables into the final model only if their *p*-value was <0.05. The excluded variables did not meaningfully impact the estimated odds ratios of the variables retained in the final model. The models adjusted for number of HIs, age at first HI (<18, 18–39, 40+), loss of consciousness (none, less than 5 min, less than one hour, less than one day, longer than one day), emergency department (ED) or hospitalization from HI (yes/no), skull fracture (yes/no), seizure history from HI (yes/no), memory loss (yes/no), sex (male/female), race, ethnicity (Non-Hispanic/Hispanic), education (high-school or less, some college or trade school degree, Bachelor’s degree or more), vigorous physical activity, military history, and BMI at age 40 years (below/ideal weight, overweight/obese). Categorical variables were assessed with Chi-square tests. Statistical significance was considered at 0.05. We performed all data analysis using SAS 9.4 [45].

## 3. Results

From the start of the HI survey, 5866 adults (53% of those who had enrolled) completed at least one of the Registry’s 18 surveys. Of these individuals, 3424 (58%) completed the head trauma survey and the basic demographic survey. Of those who responded to both, 1937 (56.6%) reported a history of HI. Table 1 lists the demographic characteristics of these 3424 patients, stratified by HI. Some 42.6% of respondents with a history of HI and 38.1% of those with no HI were diagnosed with ALS before the age of 60 years (*p* < 0.001). Persons with history of HI were statistically more likely to be males (60.5%) compared with those without HI (51.5%) (*p* < 0.001). ALS patients with HI exposure were also more likely to be White (97.3% vs. 95.3%, *p* = 0.008).

Patients with HI were more likely to be overweight/obese at age 40 years and at time of registration than the patients without HI (age 40 years: *p* = 0.03, at time of registration: *p* = 0.002) (Table 1). Military history was not significantly different for those reporting HI (*p* = 0.66). Respondents with HI were more likely to engage in vigorous leisure physical activity compared with those without HI exposure (88.9% vs. 84.0%, *p* < 0.001) (Table 1).

Table 2 shows the association of HI events (history of head trauma, number of injuries, and complications) with an ALS diagnosis before the age of 60 years. Patients with HI, either before or after ALS diagnosis, had 22% higher odds of developing ALS before the age of 60 years than those without HI (aOR = 1.22, 95% CI: 1.06–1.41, *p* = 0.007). However, via a sensitivity analysis, the adjusted odds ratio of 1.22 could be explained away by an unmeasured confounder that was associated with both the head injury and the ALS diagnosis before the age of 60 by an odds ratio of 1.7-fold each, above and beyond the measured confounders, but weaker confounding could not do so. The odds of a diagnosis before age 60 years increased monotonically with the number of HIs. In the crude and adjusted models, patients with one or two injuries were not statistically different from patients with no HI (aOR = 1.13, 95% CI: 0.96–1.33, *p* = 0.14). With five or more injuries, there was a 53% higher odds of diagnosis before age 60 years (aOR = 1.53, 95% CI: 1.11–2.00, *p* = 0.002). In the crude model, those patients with a HI before age 18 years were twice as likely to be diagnosed before age 60 years than those who experienced HI as an adult (aOR = 2.03, 95% CI: 1.53–2.70, *p* < 0.001). Patients who experienced HI between the ages of 18 and 39 years also had a higher odds of being diagnosed with ALS before age 60 years compared with those with HI at or after age 40 (aOR = 1.48, 95% CI: 1.06–2.06, *p* = 0.02). When examining complications from HI, we found that loss of consciousness and skull fracture were statistically significant in the crude models, but only loss of consciousness remained statistically significant after adjustment (aOR = 1.44, 95% CI: 1.18–1.75, *p* < 0.001).

When stratifying by loss of consciousness duration, less than five minutes was the only category that was statistically associated with ALS being diagnosed before age 60 years (in the crude model, aOR = 1.62, 95% CI: 1.27–2.06, *p* < 0.001) compared with those without loss of consciousness.

Table 3 compares the respondents who experienced a HI before age 18 years to those who never had any HI, since this age group had the highest adjusted odds of any age group. Age 64 years was the median age at diagnosis of ALS for respondents without HI. White, Non-Hispanic, and male respondents were more likely to have a HI exposure before age 18 years than the respondents without HI (race: *p* = 0.001, ethnicity: *p* = 0.01, sex: *p* < 0.001). Vigorous leisure physical activity and a high BMI (overweight or obese) were also associated with HI exposure before age 18 years (physical activity: *p* < 0.001, BMI at age 40: *p* < 0.001, BMI at registration: *p* < 0.001).

Table 4 shows the crude and adjusted odds ratios for respondents with an ALS diagnosis before age 60 years for patients with at least one HI prior to the age of 18 years. In the crude model, patients with HI before age 18 years had an almost 50% higher odds of developing ALS before age 60 years (aOR = 1.45, 95% CI: 1.23–1.71, *p* < 0.001) compared with ALS patients without HI. In Table 2, the number of HIs was not significant until the individual had three or more HIs. However, when we examined only the respondents with HIs before age 18 years, those respondents with one or two HIs, with the first before age 18 years, had an almost 30% higher odds of being diagnosed before age 60 years compared with the respondents with no HIs (in the adjusted model, aOR = 1.28, 95% CI: 1.05–1.56, *p* = 0.015). The odds of a diagnosis before age 60 increased further among those with five or more HIs (in the adjusted model, aOR = 1.70, 95% CI: 1.28–2.26, *p* < 0.001) for patients with HI before age 18. Respondents with HIs before age 18 years reporting complications such as loss of consciousness, ED visit or hospitalization, and skull fracture were statistically significant in the crude models to have an ALS diagnosis before age 60. After adjustments, only loss of consciousness and a visit to the ED or hospital remained statistically significant. When stratifying by length of time of unconsciousness, again, less than five minutes was the only statistically significant category for respondents with HI before age 18 years who were diagnosed with ALS before age 60 years (in the adjusted model, aOR = 1.83, 95% CI: 1.37–2.44, *p* < 0.001) when compared with patients without loss of consciousness. When considering all ages for HI exposure requiring an ED or hospitalization, the adjusted odds of being diagnosed with ALS before age 60 years (Table 2) were only slightly elevated and not statistically significant (aOR = 1.13, 95% CI: 0.96–1.32, *p* = 0.13). However, for those with HI requiring ED or hospitalization before 18 years, the odds remained significant after adjustment (in the adjusted model, aOR = 1.50, 95% CI: 1.21–1.86, *p* < 0.001). For all four mechanisms of HI (car crash, fall, fight, and explosion), there was a strong association between age at time of first HI and ALS diagnosis before age 60 years (Figure 1). Respondents with their first HI exposure from a car crash before age 18 years had the odds of an ALS diagnosis before age 60 years of 1.48 (95% CI: 1.20–1.82) times that of the respondents with no head injury (from any mechanism). A first HI from a car crash at or after age 40 years was not associated with a higher odds ratio of ALS diagnosis before age 60 years (OR = 0.87, 95% CI: 0.33–1.39) when compared with individuals without a head injury. We observed a similar pattern for HI from falls and fights. The odds of being diagnosed with ALS before age 60 years for respondents with HI by explosion remained constant (around the null value), regardless of the age at the time of the explosion (Figure 1).

## 4. Discussion

The relationship between HI and ALS has been researched previously, with mixed results. Our analyses of more than 3400 survey respondents support the association between HI before age 18 years and the onset of ALS before age 60 years. This is a novel and important finding and needs to be investigated further. This relationship contrasts with the “reverse causation” noted with later experiences of HI, occurring after age 40 years, occurring relatively close to the time of ALS diagnosis, which may be ascribed to effects of, rather than the cause of ALS.

Multiple biological mechanisms could explain the role of head injuries in ALS development. TDP-43 is a protein that has been implicated in neurodegeneration, TBIs, and ALS pathology [13]. Patients with traumatic brain injuries have been found to have an overexpression of this protein, which contributes to the neurocognitive dysfunction seen in these patients [46,47,48]. TDP-43 aggregates have also been associated with approximately 97% of ALS cases [49]. It is possible that repeated head injuries can result in the long-term accumulation of TDP-43 aggregates, which can create the neurotoxic effects that increase the risk of ALS [50]. Additionally, TBIs and head traumas are associated with neuroinflammation, which can persist for decades following the initial incident [51,52,53]. Neuroinflammation has also been observed in patients with ALS, in which inflammatory microglial hyperactivity in these patients has been linked to disease progression [50,54]. Thus, the continual neuronal damage generated from a head injury could put patients at a higher risk of developing ALS.

Existing research is inconclusive on the association between head injuries and ALS development. For example, in 2015, Fournier et al. conducted a study of 100 ALS cases that found that head injuries were not associated with ALS disease progression, earlier age at symptom onset, or TDP-43 in the brain [55]. Similarly, Peters et al. also found no association between ALS and head injuries, overall or occurring more than three years, but did find a significant association for head injuries that occurred less than a year before ALS diagnosis, suggesting reverse causation [56]. However, a case–control study conducted in Italy by Pupillo et al. of 458 cases and 820 controls found that head injuries were associated with a 59% odds of ALS [57]. In a case–control study of 722 ALS patients and 2268 controls in the Netherlands, Seelen et al. also found that head injuries were associated with a 95% odds of developing ALS [58]. Other case–control studies have reached similar conclusions [59,60]. Differences across these studies may be due to the methodology and study limitations such as small sample sizes, recall and interview bias, and the inability to investigate the head injuries in detail or account for the time the head injury took place. Some studies have reported that multiple head injuries and age at first head injury were associated with ALS, which is also consistent with the findings we observed in our study [18,20,61,62]. We report that having more than one head injury or a head injury before the age of 40 was associated with the odds of an ALS diagnosis. Collectively, these findings suggest that there are several nuances that contribute to a head injury increasing the risk of developing ALS. A single head injury event may not be the triggering event, as multiple head injuries may be required to induce significant neuroinflammation. Since the impacts of a severe head injury can be sustained for years after the incident, the effects of a head injury received at a younger age may not appear until later stages of life [51,52,53].

In this study, we also report that ALS diagnosis is associated with a loss of consciousness. However, interestingly, we found that spending less than 5 min unconscious following a head injury was also associated with ALS diagnosis, which conflicts with previous research on head injuries. It has been established that loss of consciousness and time spent unconscious following a head injury are indicative of the severity of a head injury and neurological damage [63,64,65,66]. Longer periods of unconsciousness are linked to more severe head injuries and neurological deficits [63,64,65,66]. A prospective study investigating loss of consciousness and subarachnoid hemorrhage found that almost half of the patients who were unconscious for less than 10 min scored 1 or 2 on the Hunt and Hess scale, suggesting these patients had less severe brain damage [67]. Over half of the patients who were unconscious for 10–60 min or more than 60 min scored 4 or higher on the Hunt and Hess scale, demonstrating that the longer a patient is unconscious, the more severe the neurological damage. Similarly, a study on head injuries in children in India also found that as the duration of time spent unconscious increased, so did the severity of head injury, as determined by increasing scores on the Glasgow Coma Scale [65]. It is unclear why the inverse was true among our population. Further studies are needed to assess the time spent unconscious following a head injury and ALS diagnosis.

In this study, we report that, in the adjusted models, there was no association between ALS diagnosis before age 60 years and skull fracture or between head injury before age 18 years and skull fracture. There is sparse research that has investigated the association between skull fractures and ALS. However, the results from existing research agrees with our findings. A case–control study conducted in 2013 found that there was no association between ALS risk and fractures as a result of head injury [56]. Gresham et al. also reported in their 1987 case–control study that there was no association between the development of ALS and skeletal fractures, even after taking into account the location of fractures [68]. Similarly, Williams et al. did not find an association between concussion or skull fracture and ALS risk [69]. However, a prospective study investigating the association between fractures and ALS risk found a positive association between fractures and ALS risk and that osteoporotic, non-osteoporotic, traumatic, and non-traumatic were all associated with ALS risk. When assessing the time since fracture, however, the association was significant up to 18 years, suggesting that ALS may be impacting bone health [70]. As much of this research is antiquated, further studies are required to investigate the association between head injuries, ALS risk, and skull fractures. Patients with ALS have been shown to have a higher incidence of fractures, deteriorating bone health, and lower bone density. A cross-sectional study investigating clinical markers of bone health in participants with ALS found that participants with ALS had lower bone quality compared with healthy individuals and that bone quality was poor across age groups [71]. Another cross-sectional study came to similar conclusions, reporting that patients with ALS had low bone density [72]. It has been theorized that the deterioration in bone health may be due to increased bone turnover, exposure to environmental toxins such as lead, or vitamin D deficiency [73,74,75,76]. Additionally, as the muscular system degenerates as ALS progress, the bone system gradually loses support, which can alter the stability and integrity of the bone’s structure [77,78]. Therefore, it is likely that skeletal function and health are impacted as the disease progresses, increasing the risk of fractures, and not that skull fractures increase the risk of ALS [70,71,72,79]. Further studies should investigate biomarkers of bone health in patients with ALS and the risk of fractures.

Head injuries are a growing area of public concern because of common and frequently disabling comorbidities such as cognition, vision problems, tinnitus, and pain [80]. Surprisingly, our analysis did not show a link between military history and ALS. Active-duty service members are at increased risk of TBI, and the short- and long-term morbidities related to HI in the veteran population have been a recent focus of attention [81]. Military service is a well-recognized risk factor for ALS [82]. Relatively low absolute numbers of survey respondents reporting history of military service limited the interpretation of the findings.

There were several limitations in the interpretation of this data analysis. The head and neck injury module questions did not comprehensively screen for all factors or precisely define “injury” using clinical or scientific terminology, which could permit the identification and classification of severity of TBI in this cohort. While the respondents were asked about loss of consciousness, other symptoms such as post-traumatic amnesia, alteration of mental status, or other acute physical, cognitive, or emotional sequelae following HI were not queried. That stated, we might be able to presume that the HIs recalled decades later by the survey respondents represented more serious injuries. The survey was not a random sample from the database, and it is likely that the self-identification process could have introduced some biases. Other studies identified cases through the provision of free or subsidized medicine, regardless of age or socioeconomic status [83,84,85]. Patients with Internet access were presumably more likely to participate, which could have skewed the population toward a younger, more educated patient sample. The proportion of patients younger than age 50 years (12.8%) was slightly overrepresented in our sample when compared with the National ALS Registry as a whole (10.5%) [86]. The respondents’ spatial distribution was comparable to the overall Registry, but racial diversity appeared to be underrepresented in the sample. Only 3.6% of respondents were Non-White compared with 11.6% in the Registry as a whole [86]. The lack of diversity in the sample and in the Registry limits the generalizability of the study to Non-White populations. Another possible limitation is the recall bias. Respondents were asked to enter ages and complications from HI before they were diagnosed with ALS. Some respondents might have estimated their ages and times if they did not remember the exact details. Additionally, younger individuals may have a better recall of past head traumas, especially those that occurred at a younger age, compared to older individuals. This could have impacted the age-related associations observed. We were also unable to clinically verify information regarding their HI or medical background. Considering that the survey was voluntary and not everyone who enrolled in the Registry necessarily took the survey, voluntary response bias is also possible. Additionally, genetic data were not available for the survey respondents. Approximately 10% of all ALS cases are familial or caused by genetic mutations [87]. Additional questions regarding clinical symptoms surrounding HI could better identify and stratify the severity of TBI.

## 5. Conclusions

The analyses of risk factors and the identification of etiologies in the ALS population is an ongoing effort by the National ALS Registry to reduce the burden of this disease nationally. The prevention and mitigation of diseases is the mission of CDC/ATSDR. The etiology of ALS remains unknown. This population-based data analysis of ALS patients found an overall increase in the likelihood of being diagnosed before age 60 years for patients who had a head injury before age 18 years, had three or more head injuries, or lost consciousness from a head injury. Head injuries caused by a car crash, fall, or fight before age 18 years and 18–39 years showed a higher likelihood of being diagnosed before age 60 years than the same injury types occurring at or after age 40 years. Therefore, HI risk reduction efforts, particularly in children, are imperative. Additional studies to further evaluate and build on these findings will support efforts to reduce the overall ALS risk.

## Figures and Tables

**Figure 1 brainsci-15-00143-f001:**
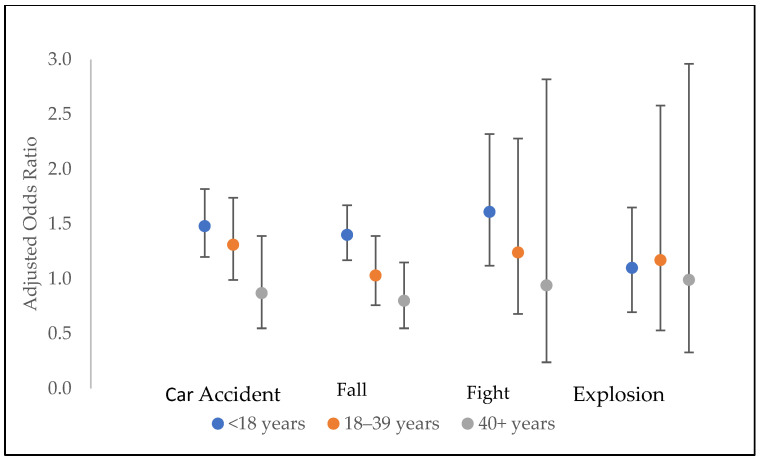
Adjusted odds ratios for ALS by age at time of first head injury relative to ALS diagnosis before age 60 years, United States, December 2014–December 2023. Patients were respondents from the National ALS Registry who completed the HI survey, n = 3424. The reference group comprised persons in the ALS Registry without head injury. Vertical bars represent 95% confidence intervals.

**Table 1 brainsci-15-00143-t001:** Demographics of ALS patients ^a^ (n = 3424) stratified by head injury (HI), National ALS Registry, United States, December 2014–December 2023.

Characteristic	No. (%)			
	Overall (N = 3424)	Patients with HI (n = 1937)	Patients Without HI (n = 1487)	*p* Value
**Age at ALS diagnosis (years)**				<0.001
18–39	101 (3.0)	53 (2.7)	48 (3.2)	
40–49	335 (9.8)	195 (10.1)	140 (9.4)	
50–59	956 (27.9)	577 (29.8)	379 (25.5)	
60–69	1336 (39.0)	758 (39.1)	578 (38.9)	
70–79	624 (18.2)	327 (16.9)	297 (20.0)	
≥80	72 (2.1)	27 (1.4)	45 (3.0)	
**Sex**				<0.001
Male	1936 (56.5)	1171 (60.5)	765 (51.5)	
Female	1488 (43.5)	766 (39.5)	722 (48.5)	
**Race ^b^**				0.008
White	3303 (96.5)	1885 (97.3)	1418 (95.3)	
Black	54 (1.6)	22 (1.1)	32 (2.2)	
Other	67 (2.0)	30 (1.6)	37 (2.5)	
**Ethnicity**				0.04
Non-Hispanic	3331 (97.3)	1894 (97.8)	1437 (96.6)	
Hispanic	93 (2.7)	43 (2.2)	50 (3.4)	
**BMI at registration**				0.002
Below/ideal weight	1150 (33.8)	607 (31.6)	543 (36.6)	
Overweight/obese	2254 (66.2)	1314 (68.4)	940 (63.4)	
**BMI at age 40**				0.03
Below/ideal weight	1105 (33.6)	597 (32.0)	508 (35.6)	
Overweight/obese	2188 (66.4)	1268 (68.0)	920 (64.4)	
**Education level**				<0.001
High-school or less	517 (15.1)	282 (14.6)	235 (15.8)	
Some college or trade school degree	772 (22.6)	486 (25.1)	286 (19.2)	
Bachelor’s degree or more	2135 (62.3)	1169 (60.3)	966 (65.0)	
**Leisure physical activity level**				<0.001
Never vigorous activity	441 (13.2)	211 (11.1)	230 (16.0)	
Vigorous activity	2893 (86.8)	1685 (88.9)	1208 (84.0)	
**Military status**				0.66
Yes	554 (16.2)	323 (16.7)	231 (15.6)	
No	2867 (83.8)	1613 (83.3)	1254 (84.4)	

^a^ ALS patients who completed the HI survey. ^b^ Other races included Asian, Native American/Alaska Native, and Unknown.

**Table 2 brainsci-15-00143-t002:** Crude and adjusted odds ratios for an ALS diagnosis before age 60 years, National ALS Registry ^a^, United States, December 2014–December 2023.

Outcome	Number (N = 3424)	Crude OR (95% CI)	*p* Value	Adjusted OR (95% CI) ^b^	*p* Value
**Head injury, any mechanism (HI)**					
Yes ^c^	1937	**1.20 (1.05, 1.38)**	**0.008**	**1.22 (1.06, 1.41)**	**0.007**
0 injuries	1487	1.00 (ref)		1.00 (ref)	
1–2 injuries	1275	1.10 (0.94, 1.28)	0.24	1.13 (0.96, 1.33)	0.14
3–4 injuries	372	**1.32 (1.05, 1.66)**	**0.02**	**1.31 (1.03, 1.67)**	**0.03**
5+ injuries	292	**1.59 (1.24, 2.05)**	**<0.001**	**1.53 (1.17, 2.00)**	**0.002**
**Age at first HI (years)**					
<18	1171	**2.29 (1.75, 2.99)**	**<0.001**	**2.03 (1.53, 2.70)**	**<0.001**
18–39	466	**1.83 (1.34, 2.48)**	**<0.001**	**1.48 (1.06, 2.06)**	**0.02**
40+	348	1.00 (ref)		1.00 (ref)	
**Loss of consciousness**					
Yes	538	**1.38 (1.15, 1.66)**	**<0.001**	**1.44 (1.18, 1.75)**	**<0.001**
**Time unconscious**					
No loss of consciousness	2721	1.00 (ref)		1.00 (ref)	
< 5 min	334	**1.56 (1.24, 1.96)**	**<0.001**	**1.62 (1.27, 2.06)**	**<0.001**
5–59 min	95	1.01 (0.66, 1.53)	0.97	1.06 (0.69, 1.63)	0.79
1–24 h	34	1.40 (0.71, 2.77)	0.33	1.65 (0.82, 3.35)	0.16
Longer than 1 day	21	1.18 (0.50, 2.82)	0.70	1.32 (0.55, 3.19)	0.54
**HI requiring emergency department or hospitalization ^c^**					
Yes	861	1.13 (0.96, 1.32)	0.13	1.15 (0.98, 1.36)	0.09
**HI causing a skull fracture ^c^**					
Yes	70	**1.75 (1.09, 2.82)**	**0.02**	1.55 (0.93, 2.60)	0.09
**HI causing a seizure ^c^**					
Yes	20	1.46 (0.61, 3.53)	0.40	1.66 (0.67, 4.12)	0.27
**HI causing memory loss ^c^**					
Yes	106	1.32 (0.89, 1.94)	0.17	1.28 (0.85, 1.91)	0.24

^a^ ALS patients who completed the HI survey. ^b^ Adjusted model, controlling for sex, race, ethnicity, education, vigorous physical activity, military history, and BMI at age 40. ^c^ Referent group was no HI. Significant associations are bolded.

**Table 3 brainsci-15-00143-t003:** Demographics of ALS patients ^a^ (n = 2456) stratified by those who experienced head injury (HI) before age 18 years and those never experiencing HI, December 2014–December 2023.

Characteristic	No. (%)		
	Patients with HI Before 18 Years (n = 1171)	Patients Without HI (n = 1487)	*p* Value
**Median age at ALS diagnosis**	59	64	0.05
**Age at diagnosis (years)**			<0.001
18–39	38 (3.2)	48 (3.2)	
40–49	133 (11.4)	140 (9.4)	
50–59	379 (32.4)	379 (25.5)	
60–69	442 (37.8)	578 (38.9)	
70–79	169 (14.4)	297 (20.0)	
≥80	10 (0.8)	45 (3.0)	
**Sex**			<0.001
Male	776 (66.3)	765 (54.5)	
Female	395 (33.7)	722 (48.5)	
**Race ^b^**			0.001
White	1147 (98.0)	1418 (95.4)	
Black	10 (0.9)	32 (2.1)	
Other	14 (1.1)	37 (2.5)	
**Ethnicity**			0.01
Non-Hispanic	1150 (98.2)	1437 (96.7)	
Hispanic	21 (1.8)	50 (3.3)	
**BMI at registration**			<0.001
Below/ideal weight	336 (28.9)	543 (36.6)	
Overweight/obese	825 (71.1)	940 (63.4)	
**BMI at age 40 years**			<0.001
Below/ideal weight	328 (29.2)	508 (35.6)	
Overweight/obese	794 (70.8)	920 (64.4)	
**Education level**			<0.001
High-school or less	146 (12.5)	235 (15.8)	
Some college or trade school degree	303 (25.9)	286 (19.2)	
Bachelor’s degree or more	722 (61.6)	966 (65.0)	
**Leisure physical activity level**			<0.001
Never vigorous activity	103 (8.9)	230 (16.0)	
Vigorous activity	1049 (91.1)	1208 (84.0)	
**Military status**			0.38
Yes	200 (17.1)	231 (15.6)	
No	971 (92.9)	1254 (84.3)	

^a^ ALS patients who completed the HI survey. ^b^ Other races included Asian, Native American/Alaska Native, and Unknown.

**Table 4 brainsci-15-00143-t004:** Crude and adjusted odds ratios for an ALS diagnosis before age 60 years for ALS patients with head injury (HI) before age 18 years compared with those diagnosed at or after age 60, National ALS Registry, December 2014–December 2023.

	Number	Crude OR (95% CI)	*p* Value	Adjusted OR (95% CI) ^a^	*p* Value
**HI**					
Yes	1171	**1.44 (1.23, 1.70)**	**<0.001**	**1.45 (1.23, 1.71)**	**<0.001**
0 injuries ^b^	1487	1.00 (ref)		1.00 (ref)	
1–2 injuries	653	**1.23 (1.02, 1.49)**	**0.03**	**1.28 (1.05, 1.56)**	**0.01**
3–4 injuries	266	**1.70 (1.31, 2.21)**	**<0.001**	**1.69 (1.29, 2.23)**	**<0.001**
5+ injuries	252	**1.79 (1.37, 2.33)**	**<0.001**	**1.70 (1.28, 2.26)**	**<0.001**
**Loss of consciousness**					
Yes/No	350	**1.48 (1.18, 1.85)**	**<0.001**	**1.55 (1.22, 1.96)**	**<0.001**
**Time unconscious**					
No loss of consciousness	2183	1.00 (ref)		1.00 (ref)	
<5 min	227	**1.74 (1.32, 2.29)**	**<0.001**	**1.83 (1.37, 2.44)**	**<0.001**
5–59 min	58	1.06 (0.62, 1.79)	0.84	1.10 (0.64, 1.90)	0.72
1–24 h	25	1.38 (0.63, 3.04)	0.42	1.66 (0.74, 3.70)	0.22
**HI requiring emergency department or hospitalization**					
Yes ^c^	453	**1.52 (1.24, 1.86)**	**<0.001**	**1.50 (1.21, 1.86)**	**<0.001**
**HI causing a skull fracture**					
Yes ^c^	43	**1.76 (0.96, 3.2)**	**0.07**	1.47 (0.77, 2.83)	0.25
**HI causing a seizure**					
Yes ^c^	11	1.15 (0.35, 3.78)	0.82	1.17 (0.35, 3.95)	0.80
**HI causing memory loss**					
Yes ^c^	60	1.06 (0.63, 1.77)	0.84	1.03 (0.61, 1.76)	0.79

^a^ Adjusted model, controlling for sex, race, ethnicity, education, vigorous physical activity, military history, and BMI at age 40. ^b^ Referent group was no HI. ^c^ Referent group was no history. Significant associations are bolded.

## Data Availability

Data are unavailable due to privacy or ethical restrictions.

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
