# Peer review of "Head Injury and Amyotrophic Lateral Sclerosis: Population-Based Study from the National ALS Registry"

_brainsci, 2025, doi:10.3390/brainsci15020143_

Round 1
Reviewer 1 Report
Comments and Suggestions for Authors
brainsci-3417464
This is an epidemiologic report addressing an important element in the pathogenesis of ALS – is a history of head trauma a clear risk factor? There is a concept that ALS may develop from a multi-event process, leading to interest in the expososome (all environmental exposures over the life of the individual). Thus a detailed exploration of prior head trauma would be valuable in the context of the expososome. It is also valuable as head trauma in athletes is considered a factor for chronic traumatic encephalopathy.
The current study is based on data from the National ALS Registry which included a module on head injury with questions about number and ages of injuries, severities (loss of consciousness, skull fracture, associated seizure, memory loss, need for medical care), circumstances (vehicular crash, fall, fight, explosion), and other possible associations such as gender, ethnicity, body mass index, participation in vigorous activities, and participation in military service.
The sample size was large, at 3,424, and 1,937 reported head trauma. Analyses were divided as those diagnosed with ALS before and after age 60 years. The findings include: 56.6% with prior head trauma, more were males and listed themselves as White, engaged in vigorous activities, but military service was not a factor. Those with head injuries were more likely to develop ALS before age 60 years, and the odds of an early diagnosis increased with number of head injuries (bur only greater than three injuries) and at earlier times of life (not age less than 18 yearsm but range 18-39 years). Loss of consciousness less than five minutes was a significant factor, but not other factors of severity. There are many other associations reported.
Factors in the development of ALS remain elusive, even in the setting of causative genes as symptoms do not develop until later in life. Epidemiologic studies of the expososome, and in particular head trauma, have not led to unequivocal factors, as discussed in this report. This study includes important details about head trauma, and questions not previously queried; one problem with more detailed questions is that there will be fewer datum points for each, affecting significance.
It is unclear what practical information to take from this study. It seems intuitive that head injury is not good, but specifics of head trauma are not a neat fit for ALS from this study. Repeated severe head trauma seems more tightly linked to chronic traumatic encephalopathy, and a PubMed search for chronic traumatic encephalopathy and ALS yielded no clearly related articles – thus the role of severe head trauma and ALS sems limited and this study provides only limited information on more mild trauma.
This study is well done, taking advantage of a large US data base and adds solid data to the literature. With respect to the expososome, it represents remote occurrences to the time of diagnosis, and does not lead to practical information as many examples of head trauma were incidental.
At the end, it is not clear what to think as there are likely biases inherent in how questions are addressed and received (who participates in the module; these issues are acknowledged by the authors). As an ALS clinician, the odds ratios seem relatively low (1.15-1.58), except for head injury before age 18 years (2.03), but it is not clear how to explain the concept of risk factors, especially when the issues are remote.
It would be helpful if the issue of chronic and severe head trauma was addressed.
Author Response
Comments 1: This is an epidemiologic report addressing an important element in the pathogenesis of ALS – is a history of head trauma a clear risk factor? There is a concept that ALS may develop from a multi-event process, leading to interest in the expososome (all environmental exposures over the life of the individual). Thus a detailed exploration of prior head trauma would be valuable in the context of the expososome. It is also valuable as head trauma in athletes is considered a factor for chronic traumatic encephalopathy.
The current study is based on data from the National ALS Registry which included a module on head injury with questions about number and ages of injuries, severities (loss of consciousness, skull fracture, associated seizure, memory loss, need for medical care), circumstances (vehicular crash, fall, fight, explosion), and other possible associations such as gender, ethnicity, body mass index, participation in vigorous activities, and participation in military service.
The sample size was large, at 3,424, and 1,937 reported head trauma. Analyses were divided as those diagnosed with ALS before and after age 60 years. The findings include: 56.6% with prior head trauma, more were males and listed themselves as White, engaged in vigorous activities, but military service was not a factor. Those with head injuries were more likely to develop ALS before age 60 years, and the odds of an early diagnosis increased with number of head injuries (bur only greater than three injuries) and at earlier times of life (not age less than 18 yearsm but range 18-39 years). Loss of consciousness less than five minutes was a significant factor, but not other factors of severity. There are many other associations reported.
Factors in the development of ALS remain elusive, even in the setting of causative genes as symptoms do not develop until later in life. Epidemiologic studies of the expososome, and in particular head trauma, have not led to unequivocal factors, as discussed in this report. This study includes important details about head trauma, and questions not previously queried; one problem with more detailed questions is that there will be fewer datum points for each, affecting significance.
It is unclear what practical information to take from this study. It seems intuitive that head injury is not good, but specifics of head trauma are not a neat fit for ALS from this study. Repeated severe head trauma seems more tightly linked to chronic traumatic encephalopathy, and a PubMed search for chronic traumatic encephalopathy and ALS yielded no clearly related articles – thus the role of severe head trauma and ALS sems limited and this study provides only limited information on more mild trauma.
This study is well done, taking advantage of a large US data base and adds solid data to the literature. With respect to the expososome, it represents remote occurrences to the time of diagnosis, and does not lead to practical information as many examples of head trauma were incidental.
At the end, it is not clear what to think as there are likely biases inherent in how questions are addressed and received (who participates in the module; these issues are acknowledged by the authors). As an ALS clinician, the odds ratios seem relatively low (1.15-1.58), except for head injury before age 18 years (2.03), but it is not clear how to explain the concept of risk factors, especially when the issues are remote.
It would be helpful if the issue of chronic and severe head trauma was addressed.
Response 1: Thank you for your assessment and review of our paper. We believe that the results from this study has practical applications for clinicians understanding potential risks factors for patients who experience head injuries. The risk factors we believe can be of particular interest is that we found an overall increase in the likelihood of being diagnosed before age 60 years for patients who had a head injury before age 18 years, had three or more head injuries, or lost consciousness from a head injury. Additionally, head injuries caused by a car crash, fall, or fight before age 18 years and 18-39 years, showed a higher likelihood of being diagnosed before age 60 years than the same injury types occurring at or after age 40 years. Although the odds ratios are somewhat low for some risk factors, they can still be of interest in a clinical setting. Additionally, these findings can be helpful for future studies that have the capability to investigate these risk factors in depth clinically or in a different study design. Unfortunately, we are unable to stratify by the severity of head traumas for this paper as we only refer to “injuries to the head or neck” in the survey. So, injuries that would be considered mild, chronic, or severe are all considered for this paper. We do believe this would be an important consideration in future studies. We discuss this as a limitation in the discussion section page 13, paragraph 7, lines 335-338:
“The head and neck injury module questions did not comprehensively screen for all factors or precisely define “injury” using clinical or scientific terminology, which could permit identification and classification of severity of TBI in this cohort.”
Reviewer 2 Report
Comments and Suggestions for Authors
This study uses data from the National ALS Registry to investigate the relationship between head injuries (HI) and the age of ALS diagnosis. Researchers found a significant association between early-life HI (before age 18) and earlier ALS diagnosis, suggesting HI may be a risk factor, particularly for childhood injuries. Multiple HIs and loss of consciousness following HI also correlated with earlier diagnosis. However, the study acknowledges limitations such as potential recall bias and sample selection bias.
A key strength is the specific examination of the association between HI before age 18 and ALS diagnosis before age 60. This focus addresses a crucial gap in the literature and provides novel insights into the potential long-term impact of early life HI on neurodegenerative disease risk. This manuscript presents a valuable contribution to the understanding of the relationship between HI and ALS. However, I do have some points to consider:
1. The study does not include any clinical verification of self-reported HIs. This raises concerns about the accuracy and reliability of the data.
2. Although the study uses adjusted models to account for potential confounders, residual confounding remains a possibility. The authors could explore sensitivity analyses to assess the robustness of their findings to unmeasured or poorly measured confounders.
3. The study reports no association between skull fracture and ALS diagnosis (similar to https://doi.org/10.3109/21678421.2012.754043). While this aligns with some prior research, further discussion is needed to clarify potential explanations for this finding, particularly in light of the known relationship between ALS progression and increased fracture risk. SAuch as "Among individual body sites, only head injury tended to be associated with a higher risk of ALS (OR=1.4, 95 percent CI: 0.8, 2.6), but the association was not statistically significant." https://doi.org/10.1093/aje/kwm153 and "The association between head injuries and ALS was statistically significant when the meta-analysis included all the 16 studies (OR 1.45, 95% CI 1.21-1.74)." https://doi.org/10.1007/s10654-017-0327-y
Need to further discuss this conflict of results.
4. The finding that spending less than 5 minutes unconscious following a HI is associated with ALS diagnosis needs further clarification, as it seems counterintuitive based on existing research on TBI severity. Further investigation and discussion are warranted.
5. The statement, "For the purposes of this paper, HI will be used as a general term to refer to an exposure consisting of any trauma to the head region," could benefit from scientific clarification. It would be helpful to define the term "trauma" more precisely, specifying whether the term includes both blunt and penetrating trauma or other forms such as chemical or thermal injury to the head region would enhance clarity.
Minor point: Some references, such as "...Parkinson’s disease.11,12 TBI...", may be interrupted. Kindly review them.
Author Response
Comments 1: The study does not include any clinical verification of self-reported HIs. This raises concerns about the accuracy and reliability of the data.
Response 1: Thank you for your comment. We agree. However, currently the ALS Registry is not able to clinically verify self-reported HIs. We have noted this as a limitation in the discussion on page 13, paragraph 7, lines 356-357:
“We are also unable to clinically verify information regarding their HI or medical background”.
Comments 2: Although the study uses adjusted models to account for potential confounders, residual confounding remains a possibility. The authors could explore sensitivity analyses to assess the robustness of their findings to unmeasured or poorly measured confounders.
Response 2: We agree. We have explored a sensitivity analysis by calculating e-values between the association of HI events and an ALS diagnosis before 60 years of age. These results have been added to the results on page 5, paragraph 3, lines 160-164:
“However, via a sensitivity analysis, the adjusted odds ratio of 1.22 could be explained away by an unmeasured confounder that was associated with both the head injury and the ALS diagnosis before age 60 by an odds ratio of 1.7-fold each, above and beyond the measured confounders, but weaker confounding could not do so.”
Comments 3:
The study reports no association between skull fracture and ALS diagnosis (similar to https://doi.org/10.3109/21678421.2012.754043). While this aligns with some prior research, further discussion is needed to clarify potential explanations for this finding, particularly in light of the known relationship between ALS progression and increased fracture risk. SAuch as "Among individual body sites, only head injury tended to be associated with a higher risk of ALS (OR=1.4, 95 percent CI: 0.8, 2.6), but the association was not statistically significant." https://doi.org/10.1093/aje/kwm153 and "The association between head injuries and ALS was statistically significant when the meta-analysis included all the 16 studies (OR 1.45, 95% CI 1.21-1.74)." https://doi.org/10.1007/s10654-017-0327-y. Need to further discuss this conflict of results.
Response 3: Thank you for the suggestion. We have revised the discussion to further discuss potential explanations for our findings in the discussion section, page 12, paragraph 5, lines 298-326:
“In this study, we report that, in adjusted models, there was no association between ALS diagnosis before age 60 years and skull fracture or between head injury before age 18 years and skull fracture. There is sparse research that investigates the association between skull fractures and ALS. Yet, the results from existing research agrees with our findings. A case-control study conducted in 2013 found that there was no association between ALS risk and fractures as a result of head injury.[59] Gresham et al. also re-ported in their 1987 case-control study that there was no association between development of ALS and skeletal fractures, even after taking into account location of fractures.[72] Similarly, Williams et al did not find an association between concussion or skull fracture and ALS risk.[73] However, a prospective study investigating the association between fractures and ALS risk found a positive association between fractures and ALS risk and that osteoporotic, non-osteoporotic, traumatic and non-traumatic all were associated with ALS risk. Yet, when assessing time since fracture, the association was significant up to 18 years, suggesting that ALS may be impacting bone health.[74] As much of this research is antiquated, further studies are required to investigate the association between head injuries, ALS risk, and skull fractures. Patients with ALS have been shown to have a higher incidence of fractures, and deteriorating bone health, and lower bone density. A cross-sectional study investigating clinical markers of bone health in participants with ALS found that participants with ALS had lower bone quality compared to healthy individuals and that bone quality was poor across age groups.[75] Another cross-sectional study came to similar conclusions, reporting that patients with ALS had low bone density.[76] It’s been theorized that the deterioration in bone health may be due to increased bone turnover, exposure to environmental toxins such as lead, or vitamin D deficiency.[77-80] Additionally, as the muscular system de-generates as ALS progress, the bone system gradually loses support, which can alter the stability and integrity of the bone’s structure.[81,82] Therefore, it is likely that skeletal function and health is impacted as the disease progresses, increasing the risk of fractures, and not that skull fractures increase the risk of ALS.[74-76, 83] Further studies should investigate biomarkers of bone health in patients with ALS and risk of fractures.”
Comments 4: The finding that spending less than 5 minutes unconscious following a HI is associated with ALS diagnosis needs further clarification, as it seems counterintuitive based on existing research on TBI severity. Further investigation and discussion are warranted.
Response 4: Thank you for your comment. We agree and have further discussed existing research on this in the discussion section, pages 11-12, paragraph 4, lines 281-297:
“In this study, we also report that ALS diagnosis is associated with loss of consciousness. Yet, interestingly, we found that spending less than 5 minutes unconscious following a head injury was also associated with ALS diagnosis, which conflicts with previous research on head injuries. It has been established that loss of consciousness and time spent unconscious following a head injury is indicative of the severity of a head injury and neurological damage. [67-70] Longer periods of unconsciousness are linked to more severe head injuries and neurological deficits. [67-70] A prospective study investigating loss of consciousness and subarachnoid hemorrhage found that almost half of patients who were unconscious for less than 10 minutes scored 1 or 2 on the Hunt and Hess Scale, suggesting these patients had a less severe brain damage.[71] Over half of patients who were unconscious for 10-60 minutes or more than 60 minutes scored 4 or higher on the Hunt and Hess scale, demonstrating that the longer a patient is unconscious the more severe the neurological damage. Similarly, a study on head injuries in children in India also found that as the duration of time spent unconscious increased, so did the severity of head injury as determined by increasing scores on the Glasgow Coma Scale.[69] It is unclear why the inverse is true among our population. Further studies are needed to assess time spent unconscious following a head injury and ALS diagnosis.”
Comments 5: The statement, "For the purposes of this paper, HI will be used as a general term to refer to an exposure consisting of any trauma to the head region," could benefit from scientific clarification. It would be helpful to define the term "trauma" more precisely, specifying whether the term includes both blunt and penetrating trauma or other forms such as chemical or thermal injury to the head region would enhance clarity.
Response 5: Thank you for the suggestion. While it would be helpful to specifically define “trauma” using a more clinical and scientific definition, we are required to use similar wording that is used in the survey, based on the survey question “Have you ever had an injury to your head or neck?”. We have added a clarifying sentence on how head trauma is determined in the materials and methods section, page 3, paragraph 3, lines 97-99:
“Participants are determined to have head trauma via the following question: “Have you ever had an injury to your head or neck?”.”
Additionally, we acknowledge this issue in the limitations section of the discussion, page 13, paragraph 7, lines 336-337:
“The head and neck injury module questions did not comprehensively screen for all factors or precisely define “injury” using clinical or scientific terminology, which could permit identification and classification of severity of TBI in this cohort.”.
Comments 6: Minor point: Some references, such as "...Parkinson’s disease.11,12 TBI...", may be interrupted. Kindly review them.
Response 6: Thank you for pointing this out. We have corrected this error on paragraph 2, page 2, line 46: “…Parkinson’s disease.[11,12] TBI…”.
Round 2
Reviewer 2 Report
Comments and Suggestions for Authors
The authors have accurately addressed my suggestions. I have no additional comments at this time. Thank you kindly.